# Wound Healing-Promoting and Melanogenesis-Inhibiting Activities of *Angelica polymorpha* Maxim. Flower Absolute In Vitro and Its Chemical Composition

**DOI:** 10.3390/molecules26206172

**Published:** 2021-10-13

**Authors:** Su-Yeon Lee, Kyung-Jong Won, Do-Yoon Kim, Mi-Jung Kim, Yu-Rim Won, Nan-Young Kim, Hwan-Myung Lee

**Affiliations:** 1Division of Cosmetic and Biotechnology, College of Life and Health Sciences, Hoseo University, Asan 31499, Korea; tndus1735@naver.com (S.-Y.L.); doyoon@hoseo.edu (D.-Y.K.); k7716708@naver.com (M.-J.K.); bb_bb22@naver.com (Y.-R.W.); rlasksdud01@naver.com (N.-Y.K.); 2Department of Physiology and Medical Science, School of Medicine, Konkuk University, Seoul 05029, Korea; kjwon@kku.ac.kr

**Keywords:** *Angelica polymorpha* Maxim., absolute, skin wound healing, melanin biosynthesis, keratinocytes, melanoma cells

## Abstract

*Angelica polymorpha* Maxim. (APM) is used in traditional medicine to treat chronic gastritis, rheumatic pain, and duodenal bulbar ulcers. However, it is not known whether APM has epidermis-associated biological activities. Here, we investigated the effects of APM flower absolute (APMFAb) on responses associated with skin wound healing and whitening using epidermal cells. APMFAb was obtained by solvent extraction and its composition was analyzed by GC/MS. Water-soluble tetrazolium salt, 5-bromo-2′-deoxyuridine incorporation, Boyden chamber, sprouting, and enzyme-linked immunosorbent assays and immunoblotting were used to examine the effects of APMFAb on HaCaT keratinocytes and B16BL6 melanoma cells. APMFAb contained five compounds and induced keratinocyte migration, proliferation, and type IV collagen synthesis. APMFAb also induced the phosphorylations of ERK1/2, JNK, p38 mitogen-activated protein kinase, and AKT in keratinocytes. In addition, APMFAb decreased serum-induced B16BL6 cell proliferation and inhibited tyrosinase expression, melanin contents, and microphthalmia-associated transcription factor expression in α-melanocyte-stimulating hormone-stimulated B16BL6 cells. These findings demonstrate that APMFAb has beneficial effects on skin wound healing by promoting the proliferation, migration, and collagen synthesis of keratinocytes and on skin whitening by inhibiting melanin synthesis in melanoma cells. Therefore, we suggest that APMFAb has potential use as a wound healing and skin whitening agent.

## 1. Introduction

Skin wound healing is a complex process and comprised of inflammation, proliferation, and remodeling phases [1]. The skin healing process involves interactions between multiple factors such as different cell types in skin, growth factors, cytokines, and extracellular matrix (ECM) [1,2]. The occurrence of a skin wound causes keratinocytes, which make up most of the epidermis, to proliferate and migrate to the damaged area to facilitate re-epithelialization [1]. The migratory and proliferative activities of keratinocytes are activated by epidermal growth factor (EGF) and other growth factors. Collagen, a key ECM protein, can be synthesized by keratinocytes or fibroblasts and is a component of skin basement membrane [3]. Collagen is involved in all phases of wound healing and plays important roles during the proliferative and remodeling phases of wound healing [1]. The cleavage of collagen is closely associated with the activation of keratinocyte migration and proliferation [1]. When the wound healing process is abnormal or hampered, healing may be delayed, and in severe cases may not be possible. To overcome these problems, an effective safe therapeutic agent is required, and in this context, the promotion of wound healing by enhancing the migratory, proliferative, and collagen-synthesizing abilities of keratinocytes offers a therapeutic strategy for complete wound healing.

Skin pigmentation is commonly found abnormally after skin injuries such as burns, wound, or laser surgery, and during wound healing responses [4]. Abnormal pigmentation in the skin can be caused by abnormal melanin synthesis or deposition [5]. Epidermal melanin is the main determinant of skin color and protects skin cells from a variety of types of damage, including ultraviolet (UV) lights [6]. Melanin biosynthesis is induced via a multistage biochemical pathway in epidermal melanocytes that is activated by various external or internal stimulating factors such as UV radiation, drugs, or α-melanocyte-stimulating hormone (MSH) [6]. Tyrosinase, tyrosinase-related protein-1 (TRP-1), and TRP-2 are proteins that play key roles in the regulation of melanogenesis in melanocytes, and their expressions are mediated by microphthalmia-associated transcription factor (MITF), a principal regulator of melanogenesis [5]. Tyrosinase catalyzes the hydroxylation of L-tyrosine to 3,4-dihydroxy-L-phenylalanine (L-DOPA) and the oxidation of L-DOPA to L-DOPA quinone, which is converted to DOPA chrome [7]. TRP-2 catalyzes the formation of 5,6-dihydroxyindole-2-carboxylic acid (DHICA) from DOPA chrome, and TRP-1 oxidizes DHICA to produce indole-2-carboxylic acid (IQCA) [7]. However, excessive melanin synthesis and deposition in skin can cause abnormal skin pigmenting conditions such as melanoma, freckles, solar lentigo, or malignant melanoma [5]. Many anti-melanogenic agents have been developed to prevent or treat abnormal pigmentation and to whiten skin [6]. However, the agents developed to date are cytotoxic or have side effects such as erythema or dermatitis [6], and thus, safer, more effective whitening and depigmenting agents without side effects are required.

*Angelica polymorpha* Maxim. (APM; family *Umbelliferae*) is distributed in Korea, China, Japan, and New Zealand [8], and its roots are used in traditional medicine to treat or relieve chronic gastritis, stomach aches, rheumatic pain, gastric ulcers, and duodenal bulbar ulcers [9]. It has been reported that extracts of Angelina species have the ability to induce proliferation of keratinocytes or inhibit melanin synthesis in melanocytes [10,11]. APM root extract and bisabolangelone, a compound found in its root extract, have been known to have anti-ulcer effects in vivo [9] and inhibit the activities of human SH-SY5Y neuroblastoma cells [8]. However, no previous study has investigated the skin wound healing and whitening activities of APM or its crude extract. In the present study, we investigated the effects of APM flower absolute (APMFAb) on skin wound healing- and whitening-related events in vitro using human immortalized keratinocytes (HaCaT cells) and melanoma cells (B16BL6 cells), respectively, and the mechanisms involved.

## 2. Results and Discussion

### 2.1. Effects of APMFAb on the Migratory and Proliferative Activities of HaCaT Cells

To determine whether APMFAb affects skin wound healing, we investigated its effects on keratinocyte migration and proliferation, which are known to play important roles in the healing process [12]. Initially, we evaluated the effect of APMFAb (0.1–250 μg/mL) on HaCaT cell viability using a water-soluble tetrazolium salt (WST) assay. Treatment with APMFAb (0.1–150 μg/mL) had no cytotoxic effect on HaCaT cells, but rather accelerated cell viability significantly from 1 μg/mL to 100 μg/mL as compared with untreated controls. However, at 250 μg/mL APMFAb reduced HaCaT cell viabilty (Figure 1a). Cell migration was assessed using a Boyden chamber assay on HaCaT cells exposed to APMFAb (0.1–150 μg/mL), and APMFAb was found to increase migration significantly at 50 μg/mL (122.33 ± 6.33% of untreated control) (Figure 1b), indicating that APMFAb can stimulate the migration in HaCaT cells. We also investigated the effect of APMFAb (0.1–150 μg/mL) on the proliferation of HaCaT cells using a BrdU incorporation assay. The results obtained showed APMFAb at 1–50 μg/mL significantly increased HaCaT cell proliferation and increased it maximally at 10 μg/mL (159.62 ± 7.75% of untreated control) (Figure 1c), which showed that APMFAb can promote HaCaT cell proliferation.

### 2.2. Effect of APMFAb on Keratinocyte Sprout Outgrowth

To simultaneously evaluate the proliferative and migratory effects of APMFAb on keratinocytes, we used a collagen sprout assay. APMFAb (1–100 μg/mL) significantly elevated HaCaT cell sprouting at 10 and 50 μg/mL, peaking at 50 μg/mL (326.24 ± 14.99% versus untreated control) (Figure 2). This assay is commonly used to confirm keratinocyte proliferation and migration results in vitro [12].

### 2.3. Effect of APMFAb on the Activations of Kinases in HaCaT Cells

To investigate the association between signaling molecules and the migration and proliferation of HaCaT cells exposed to APMFAb (0.1–150 μg/mL), we performed western blotting. APMFAb significantly elevated the activations of extracellular signal-regulated kinase1/2 (ERK1/2) from 10 μg/mL to 100 μg/mL (Figure 3a,b), c-Jun N-terminal kinase (JNK) at 10 and 50 μg/mL (Figure 3a,c), p38 mitogen-activated protein kinases (MAPK) (Figure 3a,d) at 100 and 150 μg/mL, and serine/threonine-specific protein kinase (AKT) at 10 and 50 μg/mL (Figure 3a,e). The activations of ERK1/2 and JNK peaked at an APMFAb concentration of 10 μg/mL (209.65 ± 15.47% (Figure 3b) and 189.30 ± 8.87% (Figure 3c), respectively, versus untreated controls). p38 MAPK and AKT were maximally activated at 150 (226.72 ± 5.70%; Figure 3d) and 50 μg/mL (196.35 ± 6.21; Figure 3e), respectively.

MAPKs regulate many different cellular responses besides proliferation and migration, and ERK1/2, JNK, and p38 MAPK are major members of the MAPK family [13,14]. It has been reported that MAPKs and AKT importantly signal the migration and proliferation of keratinocytes [14]. Increased activation of ERK1/2 is known to promote keratinocyte migration and proliferation [14], and reductions in ERK1/2 activation have the opposite effect [15]. Increased p38 MAPK activation also promotes keratinocyte proliferation and migration [16], whereas though JNK participates in keratinocyte migration and proliferation, its effects are consistent. For example, JNK activation has been reported to promote, and on the other hand, not to be involved in keratinocyte migration and proliferation [14,17], which implies that JNK signaling does not play a critical role in migratory and proliferative activities in keratinocytes. As mentioned above, we found that APMFAb increased the phosphorylation of ERK1/2, p38 MAPK, and JNK in HaCaT cells, which suggests activations of these MAPKs are associated with APMFAb-induced migration and proliferation in HaCaT cells. We also showed that APMFAb increased AKT activation in HaCaT cells, and AKT activation has been reported to promote HaCaT cell migration and proliferation [14]. These observations indicate APMFAb might induce keratinocyte migratory and/or proliferative activities by activating the AKT and/or MAPK signaling pathways.

### 2.4. Effect of APMFAb on Collagen Synthesis by HaCaT Cells

To examine the effect of APMFAb on collagen synthesis by keratinocytes, we performed sandwich enzyme-linked immunosorbent assay (ELISA) using conditioned medium, which was obtained by culturing HaCaT cells in the presence of APMFAb. Treatment of HaCaT cells with APMFAb from 1 μg/mL to 100 μg/mL did not affect type I collagen synthesis (Figure 4a) but significantly elevated type IV collagen synthesis at 100 μg/mL (202.87 ± 14.79% versus untreated control, Figure 4b).

Collagen synthesis is required for all skin healing processes but is especially important during the proliferation and remodeling phases [1,18]. Types I and IV collagen were found to increase keratinocyte migratory activity in vitro [18], and these two collagens are associated with skin recovery [19] and skin membrane formation [18,20], respectively. Furthermore, both collagens are produced and secreted by keratinocytes or fibroblasts exposed to various plant extracts [12]. Therefore, our findings indicate that APMFAb facilitates skin wound healing by enhancing type IV collagen production in keratinocytes.

### 2.5. Effects of APMFAb on Melanin Formation in B16BL6 Melanoma Cells

To examine the skin whitening-associated activities of APMFAb, we treated B16BL6 melanoma cells with α-MSH (200 nM) in the presence or absence of APMFAb (0.1–100 μg/mL) and analyzed melanin formation-related responses. Treatment with APMFAb significantly reduced the fetal bovine serum (FBS) (2%, *v*/*v*)-induced proliferations of B16BL6 melanoma cells at 50 and 100 μg/mL, peaking at 100 μg/mL (93.23 ± 11.19% versus 218.89 ± 17.18% [FBS (2%, *v*/*v*) alone-treated cells]; Figure 5a). In addition, APMFAb significantly attenuated the α-MSH-induced expression of MITF in melanoma cells at 100 μg/mL (114.16 ± 4.83% versus 155.89 ± 12.56% [α-MSH alone-treated cells]; Figure 5b). Furthermore, treatment of B16BL6 melanoma cell with APMFAb (0.1–100 μg/mL dose-dependently decreased α-MSH-enhanced tyrosinase expression and this response was greatest at 100 μg/mL (71.12 ± 2.45% versus 144.01 ± 0.39% [α-MSH alone-treated cells]; Figure 5c). APMFAb at 0.1–100 μg/mL also dose-dependently inhibited α-MSH-induced melanin production in B16BL6 melanoma cells, and its inhibitory effect peaked at 100 μg/mL (101.48 ± 1.54 % versus 235.78 ± 0.00% [α-MSH alone-treated cells]; Figure 5d). Furthermore, APMFAb at 0.1 and 100 μg/mL had no cytotoxic effect on B16BL6 melanoma cells (data not shown).

Melanin biosynthesis is influenced by the survival, proliferation, and differentiation of melanocytes [21] and is regulated by tyrosinase, TRP-1, and TRP-2 [22], the expressions of which were reported to be reduced by inhibiting MITF [23], implying that MITF can regulate tyrosinase, TRP-1, and TRP-2-induced effects. Furthermore, it has been shown that inhibitions of the expression of these proteins reduced melanin production in melanocytes [22], and in another study, melanin biosynthesis was attenuated by inhibiting melanocyte proliferation [23]. Similarly, we observed APMFAb inhibited the expressions of these proteins and proliferation and melanin production in B16BL6 melanoma cells. Therefore, our findings indicate APMFAb inhibits melanin biosynthesis in melanocytes.

### 2.6. Chemical Composition of APMFAb

Gas chromatography/mass spectrometry (GC/MS) analysis showed that APMFAb contained 5 compounds (Figure 6 and Table 1), viz., isopimpinellin (34.95%), bergapten (30.74%), nonadencane (24.50%), aromadendrene (8.70%), and methoxsalen (1.12%) (Table 1). It has been reported that methoxsalen, in combination with long wavelength UVA inhibits HaCaT cell migration [24]. On the other hand, isopimpinellin or bergapten-stimulated melanogenesis in melanoma cells [25,26]. Accordingly, it might be expected that APMFAb would have a negative effect on skin wound healing or whitening response. However, other compounds are not reported about their skin wound healing- or melanogenesis-linked bioactivities on keratinocytes and melanocytes. *Angelica dahurica* extract increased human keratinocyte proliferation [10] and *Angelica tenuissima* root extract inhibited melanin production in melanoma cells [11]. In the present study, we found that APMFAb promoted skin wound healing and whitening response. These observations indicate that APMFAb may contain components that promoted these responses. Further studies are required to identify the main bioactive components in APMFAb responsible for its wound healing and/or whitening-related responses.

## 3. Materials and Methods

### 3.1. Materials

Trypsin-ethylenediamine tetra-acetic acid (EDTA), FBS, and penicillin/streptomycin (P/S) were purchased from Gibco BRL (Gaithersburg, MD, USA), and phosphate buffered saline (PBS) and Dulbecco’s modified eagle medium (DMEM) were from Welgene (Daegu, South Korea). Bovine serum albumin (BSA), L-DOPA, Triton X-100, phenylmethylsulfonyl fluoride, α-MSH, and dimethyl sulfoxide (DMSO) were purchased from MilliporeSigma (St. Louis, MO, USA), and recombinant human EGF (purity > 97%) was from R&D Systems (Minneapolis, MN, USA). The EZ-CyTox kit was purchased from DoGenBio (Seoul, Korea) and type I collagen from BD Bioscience (Franklin Lakes, NJ, USA). The antibodies used were anti-ERK1/2, anti-phospho ERK1/2, anti-p38 MAPK, anti-phospho p38 MAPK, anti-JNK, anti-phospho JNK, anti-AKT, anti-phospho AKT, anti-MITF, anti-rabbit IgG, and anti-mouse IgG (Cell Signaling, Beverly, MA, USA), monoclonal anti-type I and IV collagens, polyclonal anti-type I and IV collagen (Abcam), anti-tyrosinase (Santa Cruz Biotechnology, CA, USA), and β-actin (MilliporeSigma, St. Louis, MO, USA).

### 3.2. Extraction of Angelica Polymorpha Maxim. Flower Absolute

APM flowers were collected from plants growing near Hanaro Farm, Songji-myeon, Haenam-gun, Jeollanam-do, South Korea (34°23′00.4″ N 126°33′59.0″ E; 8 October 2019) and identified by Jong-Cheol Yang, Division of Forest Biodiversity and Herbarium, Korea National Arboretum, South Korea. A voucher specimen (No. AP-0001) was kept at the Herbarium of the College of Life and Health Science, Hoseo University, South Korea. Absolute was obtained by solvent extraction, as previously described [12]. In brief, APM flowers (5.521 kg) were completely immersed in hexane at room temperature (RT) for 1 h. Extracts were collected, and the hexane was removed by rotary evaporation at 25 °C under vacuum to give a dark yellow waxy residue (concrete). This residue was then mixed with ethanol (99.5%), left at −20 °C for 12 h, filtered through a sintered funnel, and then the ethanol was removed by evaporation at 35 °C to leave a light-yellow anhydrous wax (APMFAb; 1.43 g, yield 0.026%, *w*/*w*). APMFAb was stored at −80 °C until required.

### 3.3. Identification of Compounds in APMFAb

Components in APMFAb were identified by GC/MS at the Korean Basic Science Institute (KBSI, Seoul, Korea). GC/MS analysis was executed using an Agilent 7890BGC/7010QQQ MS instrument (Palo Alto, CA, USA) and a DB5-MS capillary column (30 m × 0.25 mm, film thickness 0.25 μm) [12]. Helium was used as the carrier gas and its flow rate was 1 mL/min. Injector port, ion source, and interface temperatures were 290, 230, and 290 °C, respectively. The GC oven was programmed as follows; 40 °C for 3 min, 40 to 230 °C at 2 °C/min, 230 to 300 °C at 5 °C/min, and maintained at 300 °C for 15 min. The split ratio was 1:10. Masses were scanned from m/z 50 to 800. Retention indices (RIs) were determined using Kovats method using C_7_–C_40_ n-alkanes as standards. Compounds were identified by comparing their RI values with Kovats indices [27] and matching their MS fragmentation patterns with those in the Wiley7NIST0.5L Mass Spectral library and mass spectrum catalogs. GC/MS data were reanalyzed as described by Adams [28].

### 3.4. Cell Culture

HaCaT cells (a human immortalized keratinocyte cell line) were obtained from the National Institute of Korean Medicine Development (NIKOM, Gyeongsan, South Korea) and B16BL6 cells (a murine melanoma cell line) were from the Korean Cell Line Bank (KCLB, Seoul, South Korea). Cells were cultured in DMEM with 1% (*v*/*v*) P/S and 10% (*v*/*v*) FBS in a humidified 5% CO_2_ atmosphere at 37 °C. For experiments, cells were cultured until 70–80% confluent.

### 3.5. WST Assay

The EZ-CyTox kit (DoGenBio, Seoul, South Korea) was used to analyze the cell viability of HaCaT cells [12]. Cells (5 × 10^3^ cells/well) were seeded in 96-well microtiter plates and then treated with various concentrations of APMFAb (dissolved in DMEM containing 0.5% (*v*/*v*) DMSO) for 48 h. Subsequently, cells were loaded with EZ-CyTox reagent (10 μL) and incubated for 30 min at 37 °C. Absorbance levels were measured using a multi-well plate reader (Synergy 2, Bio-Tek Instruments, Winooski, Vermont, USA) at 450 nm.

### 3.6. BrdU Incorporation Assay

HaCaT cell and B16BL6 cell proliferations were determined using a BrdU (5-bromo-2′-deoxyuridine) incorporation assay using a cell proliferation ELISA, BrdU kit (Roche, Indianapolis, Indiana, USA) [12]. Briefly, HaCaT (3 × 10^3^ cells/well) and B16BL6 cells (2 × 10^3^ cells/well) were incubated in 96-well cell culture plates for 12 h. Thereafter, cells were treated with different concentrations of APMFAb (dissolved and diluted in DMEM containing 0.5% (*v*/*v*) DMSO) for 36 h, washed with DMEM, and fixed for 30 min at RT. After washing with PBS, cells were labeled with BrdU-labeling solution (10 μM) and incubated for 12 h at 37 °C to denature DNA. Peroxidase labeled anti-BrdU monoclonal antibody was then added to the cells, followed by incubation at RT for 90 min. Luminescence was measured using a luminometer (Synergy 2; Bio-Tek Instruments, Winooski, VT, USA).

### 3.7. Migration Assay

Cell migration was analyzed using a 48-well Boyden microchemotaxis chamber (Neuro Probe Inc., Gaithersburg, MD, USA), as previously described [9]. Briefly, lower chamber wells were loaded with DMEM containing 0.1% (*w*/*v*) BSA and different concentrations of APMFAb. A membrane coated with 0.1 mg/mL type Ι collagen was then laid over lower chamber wells, and upper chambers were loaded with cells (5 × 10^4^ cells/well) in DMEM containing 0.1% (*w*/*v*) BSA. Assembled chambers were incubated for 210 min at 37 °C, and membranes were then removed, fixed, and stained using Diff-Quick (Baxter Healthcare, Miami, FL, USA). Numbers of cells that migrated onto lower membrane surfaces were counted using an optical microscope (× 200) (Carl Zeiss, Jena, Thuringen, Germany).

### 3.8. Collagen Sprout Assay

Cell migration and proliferation were analyzed using the collagen sprout assay [12]. HaCaT cells (2.5 × 10^4^ cells/mL) were mixed with type I collagen, 10× DMEM medium, and 1 N NaOH (pH 7.2). The mixture was spotted in each well of 24-well cell culture plates. After drying for 20 min, spots were treated with or without APMFAb, incubated at 37 °C in CO_2_ incubator for 48 h, and then fixed and stained using Diff-Quik solution. The spot images were photographed using an optical microscope (Carl Zeiss, Jena, Thuringen, Germany) (×100). Lengths of sprouts were measure using Image J software (Version Java 1.8.0., NIH, Bethesda, MD, USA).

### 3.9. Collagen Synthesis Assay

Collagen synthesis was analyzed by ELISA, as previously described [12]. Briefly, HaCaT cells (5 × 10^5^ cells/well) were seeded in 100-mm cell culture dishes and incubated with different concentrations of APMFAb for 48 h at 37 °C. Media were centrifuged sequentially at 500, 800, and 1000× *g* for 10 min, and the supernatants obtained (conditioned media; 100 μL/well) were added to 96-well microtiter plates coated with capture antibody (type I or IV collagen monoclonal antibody) and then incubated with biotin-conjugated collagen type I or IV polyclonal antibody (dilution 1:2000 in 1% [*w*/*v*] BSA/ PBS) for 90 min at RT. Wells were then washed with PBS, loaded with streptavidin-horseradish peroxidase conjugate (Roche) (diluted 1:5000 in 1% [*w*/*v*] BSA/PBS), incubated for 1 h at RT, and washed with PBS. Enhanced chemiluminescence (ECL) solution (Thermo Fisher Scientific, Waltham, MA, USA) was then added, and luminescence was measured using a luminometer (Synergy 2, Bio-Tek Instruments, Winooski, VT, USA).

### 3.10. Western Blotting

Cells were lysed using radioimmunoprecipitation assay buffer (RIPA buffer; Cell Signaling), and lysates were centrifuged at 17,000× *g* for 15 min at 4 °C. Protein concentrations in supernatants were determined using DC protein assay reagents (Bio-Rad Laboratories, Hercules, CA, USA). Proteins (80–120 μg/lane) were subjected to 10% (*w*/*v*) SDS-PAGE and transferred to polyvinylidene fluoride membranes at 4 °C. Membranes were blocked in 3% (*w*/*v*) skim milk at RT for 2 h, washed with PBS containing 0.05% (*v*/*v*) Tween-20, incubated with primary antibodies (1: 1000–10,000 dilution), and then with secondary antibody (conjugated horseradish peroxidase) at RT for 1 h. Bands were visualized using a chemiluminescence substrate and detected using a chemiluminescence imaging system (LuminoGraph, ATTO, Tokyo, Japan).

### 3.11. Melanin Content Assay

Melanin contents were analyzed as described previously [29]. Briefly, B16BL6 melanoma cells (5 × 10^5^ cells per well) were incubated in DMEM containing different concentrations of APMFAb with or without 200 nM α-MSH for 48 h at 37 °C in 60-mm dishes. After washing with PBS, cells were lysed using lysis buffer (0.1 M sodium phosphate buffer [pH 6.8] containing 0.2 mM phenylmethylsulfonyl fluoride and 1% [*v*/*v*] Triton X-100 and then centrifuged at 10,000× *g* for 15 min. Cell pellets were dissolved in 150 μL of 1 N NaOH containing 10% (*v*/*v*) DMSO, incubated at 80 °C for 1 h, and pipetted to solubilize the melanin. Absorbances were measured at 405 nm using an ELISA reader (Synergy 2, Bio-Tek Instruments, Winooski, VT, USA).

### 3.12. Statistical Analysis

Data values are presented as means ± standard errors of means (SEMs). The Student’s *t*-test is used to analyze the significances of differences between pairs of groups. One-way analysis of variance (ANOVA) followed by the Tukey *post hoc* test was conducted for multiple comparisons. The analysis was conducted using GraphPad Prism (version 5.0; GraphPad Software, Inc., San Diego, CA, USA), and *p* values < 0.05 were considered significant.

## 4. Conclusions

In the present study, we found APMFAb contained five compounds and that it stimulated HaCaT cell proliferation, migration, and sprout outgrowth, and increased the activations of AKT, ERK1/2, p38 MAPK, and JNK, and increased type IV collagen synthesis in HaCaT cells. In addition, in B16BL6 cells exposed to α-MSH, APMFAb reduced serum-induced proliferation and attenuated MITF expression, tyrosinase expression, and melanin production. These observations suggest APMFAb induces wound healing-linked migration, proliferation, and collagen synthesis in keratinocytes and promotes whitening-associated responses in melanocytes. Our findings suggest that APMFAb provides a novel basis for the development of natural skin wound healing and whitening agents.

## Figures and Tables

**Figure 1 molecules-26-06172-f001:**
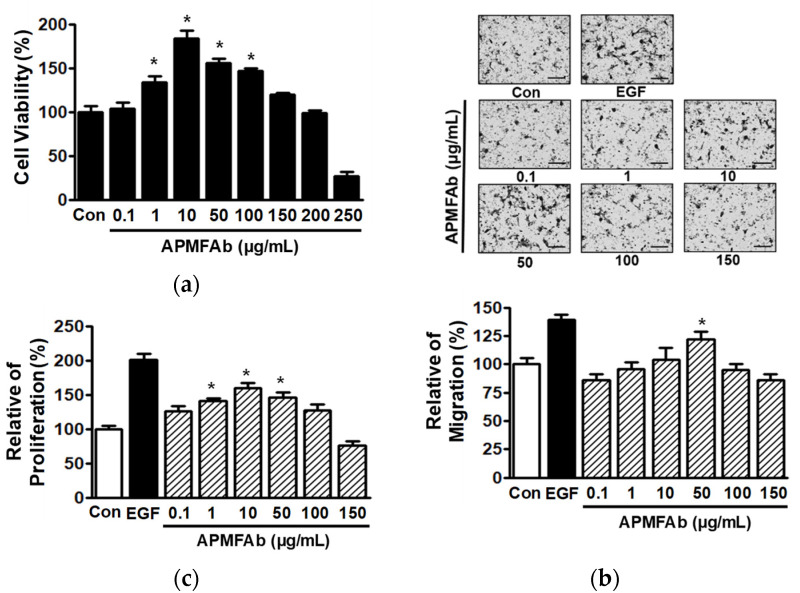
Effects of *Angelica polymorpha* Maxim. flower absolute on HaCaT cell migration and proliferation. (**a**) Cell viability. HaCaT cells were treated with *Angelica polymorpha* Maxim. flower absolute (APMFAb; 0.1–250 µg/mL) for 48 h. Viabilities were measured using the WST assay (*n* = 5). (**b**) Cell migration. HaCaT cells were exposed APMFAb (0.1–150 µg/mL) for 210 min. Migration levels were determined using a Boyden chamber assay as described in Materials and Methods (upper panels). Representative images of the results. Black spots indicate migrated cells. Scale bar = 100 μm. (lower panels) Graphical representation of the results. Recombinant human epidermal growth factor (EGF: 1 ng/mL) was used as the positive control. Results are presented as mean percentages ± SEMs of non-treated controls (Con) (*n* = 4). * *p* < 0.05 vs. non-treated cells. (**c**) Cell proliferation. HaCaT cells were incubated with APMFAb (0.1–150 µg/mL) for 48 h, and proliferations were determined using a BrdU incorporation assay as described in Materials and Methods. Recombinant human epidermal growth factor (EGF: 50 ng/mL) was used as the positive control. Results are presented as mean percentages ± SEMs of non-treated controls (Con) (*n* = 5). * *p* < 0.05 vs. non-treated cells.

**Figure 2 molecules-26-06172-f002:**
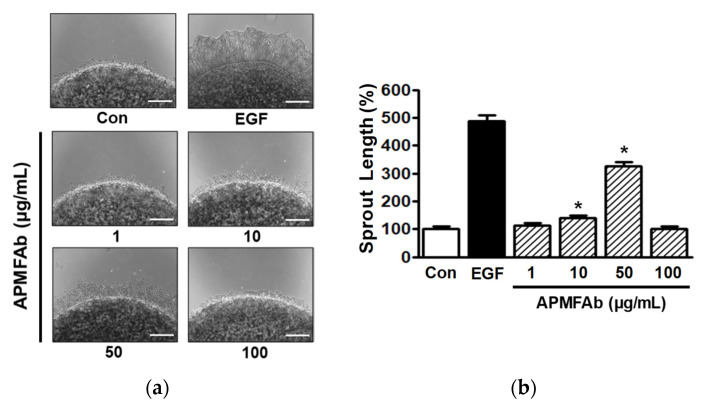
Effect of *Angelica polymorpha* Maxim. flower absolute on sprout formation by HaCaT cells. (**a**) HaCaT cells were mixed with collagen and spotted on 24-well plate. Spots were incubated with or without *Angelica polymorpha* Maxim. flower absolute (APMFAb) (1–100 µg/mL) for 48 h. Spots and sprouts were then stained with Diff-Quick solution and images were acquired using a microscope. Epidermal growth factor (EGF) (50 ng/mL) was used as the positive control. Scale bar = 100 μm. (**b**) Statistical graph obtained from panel A. Results are presented as mean percentages ± SEMs of non-treated controls (Con) (*n* = 3). * *p* < 0.05 vs. non-treated controls.

**Figure 3 molecules-26-06172-f003:**
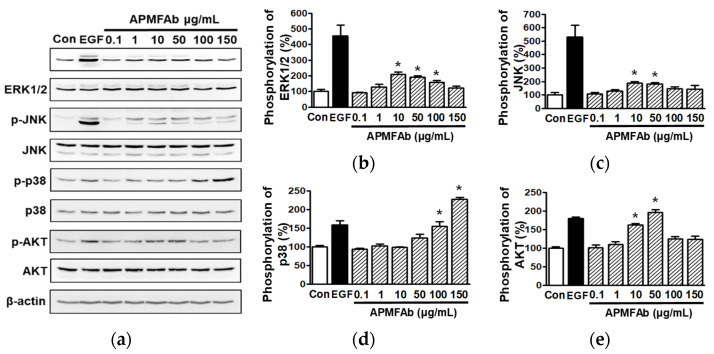
Effect of *Angelica polymorpha* Maxim. flower absolute on the phosphorylation of kinases in HaCaT cells. HaCaT cells were treated with or without APMFAb (0.1–150 µg/mL) for 10 min. Cell lysates were immunoblotted with antibodies of each kinase or β-actin. (**a**) Representative images. (**b**–**e**) Graphical representation of phosphorylated ERK1/2 (**b**), JNK (**c**), p38 MAPK (**d**), and AKT expression levels (**e**) shown in panel (**a**). Kinase phosphorylation levels are expressed as percentages of levels in non-treated controls (Con). Recombinant human epidermal growth factor (EGF: 5 ng/mL) was used as the positive control. Results are presented as means ± SEMs (*n* = 3/protein). * *p* < 0.05 vs. non-treated cells. p-ERK 1/2, phosphorylated ERK 1/2; p-JNK, phosphorylated JNK; p-p38, phosphorylated p38 MAPK; p-AKT, phosphorylated AKT.

**Figure 4 molecules-26-06172-f004:**
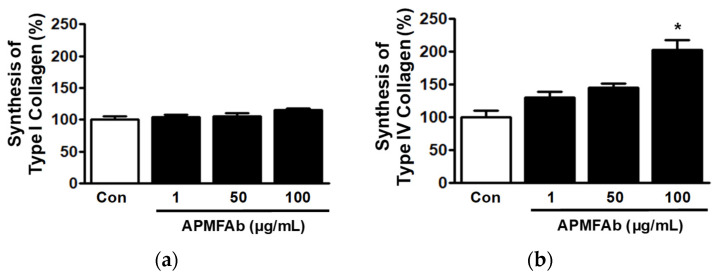
Effects of *Angelica polymorpha* Maxim. flower absolute on the syntheses of type I and IV collagens. HaCaT cells were incubated in the absence or presence of APMFAb (1–100 µg/mL) for 48 h. Conditioned media were subjected to sandwich ELISA using anti-type I (*n* = 3; **a**) or anti-type IV collagen antibody (*n* = 3; **b**). Collagen levels in conditioned media are expressed as percentages of those in non-treated controls (Con). Results are presented as means ± SEMs. * *p* < 0.05 vs. non-treated cells.

**Figure 5 molecules-26-06172-f005:**
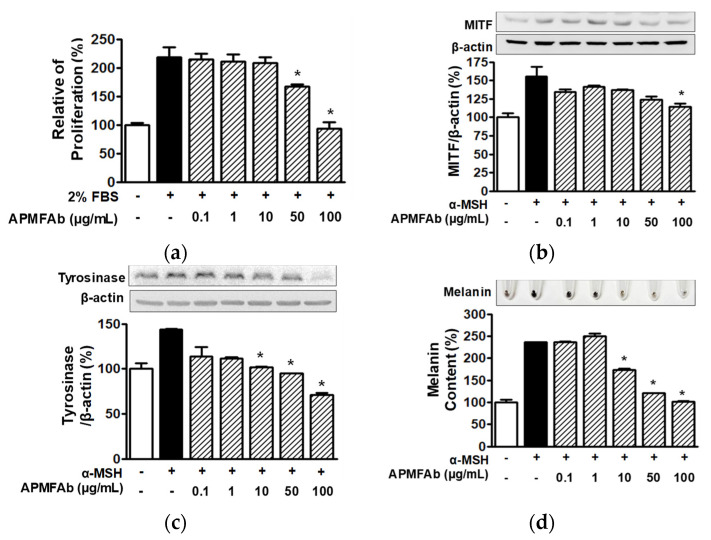
Effects of *Angelica polymorpha* Maxim. flower absolute on melanogenesis-associated responses in B16BL6 melanoma cells. (**a**) Cell proliferation. B16BL6 melanoma cells were incubated in DMEM in the presence or absence of 2% (*v*/*v*) FBS with or without APMFAb (0.1–100 μg/mL) for 48 h at 37 °C. Cell proliferations were determined using the BrdU incorporation assay as described in Materials and Methods (*n* = 5). Results are presented as percentages of response in the quiescent state. * *p* < 0.05 vs. cells treated with 2% (*v*/*v*) FBS alone. (**b**–**d**) Melanogenesis-linked responses in B16BL6 melanoma cells. B16BL6 melanoma cells were pretreated with or without DMEM containing a-MSH (200 nM) for 10 min and then incubated with DMEM containing different concentrations of APMFAb (0.1–100 μg/mL) in the presence or absence of 200 nM α-MSH for 5 min. Cells were analyzed for MITF (*n* = 3; **b**) and tyrosinase expressions (*n* = 3; **c**) by Western blot. Melanin content analysis (*n* = 4; **d**) was performed as described in Materials and Methods. Upper images in panels (**b**–**d**) show representative responses. Results are presented as mean percentages ± SEMs of mean response in the quiescent state. * *p* < 0.05 vs. cell treated with α-MSH alone.

**Figure 6 molecules-26-06172-f006:**
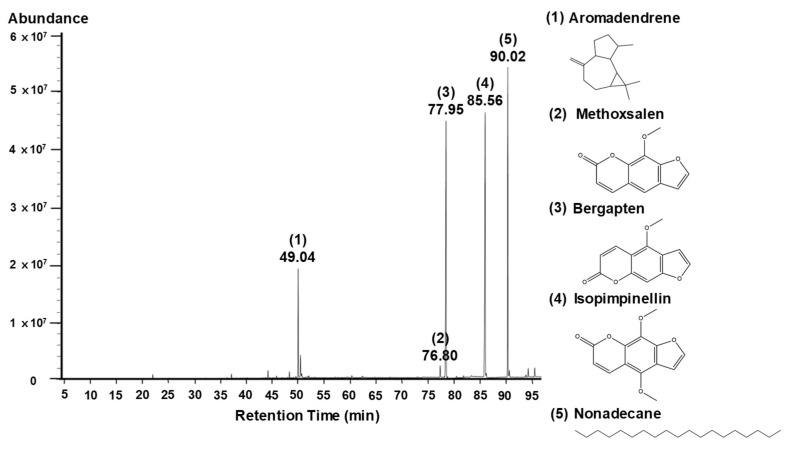
GC/MS total ion chromatogram of *Angelica polymorpha* Maxim. flower absolute. Peak numbers indicate compound numbers (bracketed); retention times (below bracketed numbers) of the 5 identified compounds are listed in Table 1. The chemical structures of the 5 compounds are shown on the right of the chromatogram.

**Table 1 molecules-26-06172-t001:** Chemical compounds identified in the GC/MS assay of *Angelica polymorpha* Maxim. flower absolute.

No	Component Name	RT ^1^	RI ^2^		Area (%)	CAS No.
			Observed	Literature		
1	Aromadendrene	49.04	1479	1479	8.70	489-39-4
2	Methoxsalen	76.80	2000	2009	1.12	298-81-7
3	Bergapten	77.95	2025	2025	30.74	484-20-8
4	Isopimpinellin	85.56	2195	2195	34.95	482-27-9
5	Nonadencane	90.02	2299	2299	24.50	629-92-5
Total Identified (%)					100.00	

^1^ RT: Retention time, ^2^ RI: Retention indices determined using a DB-5MS capillary column.

## Data Availability

Not applicable.

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
