# Peer review of "Wound Healing-Promoting and Melanogenesis-Inhibiting Activities of Angelica polymorpha Maxim. Flower Absolute In Vitro and Its Chemical Composition"

_molecules, 2021, doi:10.3390/molecules26206172_

Round 1
Reviewer 1 Report
The manuscript only has some typing errors, which were highlighted in yellow in the PDF file.

Author Response
The manuscript only has some typing errors, which were highlighted in yellow in the PDF file.
(Response) Thank you. Based on the reviewer comments, we revised our manuscript (commented some typing errors: Angelica Polymorpha Maxim, in vitro, Absolute was extracted by solvent extraction, 95% air/5% CO2, 122.33±6.33%, etc.)

Reviewer 2 Report
This paper describes the activity of an extract of flowers of Angelica Polymorpha Maxim on the healing of the skin and its whitening.
- The roots of this plant are commonly used in traditional medicine against chronic gastritis, stomach aches, rheumatic pain, gastric ulcers, and duodenal bulbar ulcers. They inhibit the activities of human SH-SY5Y neuroblastoma cells. Why do the authors believe that this extract exhibits activity on healing of the skin and its whitening?
- In the introduction, the transition between the two supposed effects of the extract is brutal (lines 48-50).
- Lines 86/123/160: “from 1/10 µg / mL to 100 µg / mL” instead “at 1/10 to 100”?
- The concentration is in µg of dry residue / mL of solvent?
- Line 86. The authors say that the concentration of 250 µg / mL is toxic for the cell (Fig.1A). The results of this toxicity are not shown.
- Line 100: "Violet" spots. The sent paper is in black and white
- Lines 98 and 104: “Viabilities” and “Recombinant” instead of “Via-bilities” and “Recom-binant”?
- Line 125 "at 10 and 50 µg / mL" instead of “at 10 to 50"
- Lines 157-174: the authors demonstrate that “the extract elevated type IV collagen synthesis and did not affect type I collagen synthesis”. Can the authors discuss this point? Why collagen IV and not collagne I?
- - lines 214-215. The % are obtained by internal standardization?
100 * (Peak area) / (Total area)
- Lines 212-230: The discussion on the correlation between the composition of the extract and its activity is too short. Moreover, for certain molecules, contrary effects are described in the literature. At what concentrations did these effects occur? The authors conclude that there may be interactions between molecules.
Author Response
This paper describes the activity of an extract of flowers of Angelica polymorpha Maxim. on the healing of the skin and its whitening.
Comment 1. The roots of this plant are commonly used in traditional medicine against chronic gastritis, stomach aches, rheumatic pain, gastric ulcers, and duodenal bulbar ulcers. They inhibit the activities of human SH-SY5Y neuroblastoma cells. Why do the authors believe that this extract exhibits activity on healing of the skin and its whitening?
(Response) In the present study, we found that Angelica Polymorpha Maxim. flower absolute (APMFAb) promoted proliferation, migration, and type IV collagen synthesis in keratinocytes. As mentioned in our manuscript, keratinocyte is a main cell of the epidermis and the ability of proliferation and migration of keratinocytes can play a critical role in promotion of wound healing/skin regeneration [1]. Our results indicate that APMFA facilitates proliferation and migration of keratinocytes. Skin damage causes the keratinocytes to initiate a repair process in contact with collagen and to move across the wound layer [1]. Collagen synthesis is an important response that promotes skin pro-inflammatory activity and epithelialization to skin regeneration in injured skin [2]. collagen type IV synthesis plays a key role in membrane formation of skin [3, 4]. Our result imply that APMFAb can induce production of these type IV collagens in keratinocytes. Therefore, APMFAb may promote positive effects on skin wound healing events.
As described in the Introduction section of our manuscript, melanin biosynthesis is induced via a multistage biochemical pathway in epidermal melanocytes [5]. Tyrosinases are proteins that play key roles in the regulation of melanogenesis in melanocytes, and their expressions are mediated by microphthalmia-associated transcription factor (MITF), a principal regulator of melanogenesis [6]. In the present study, we found that APMFAb decreased proliferation, MITF expression, tyrosinase expression, melanin synthesis in melanocytes. Therefore, our results imply that APMFAb may promote skin whitening.
From observations from keratinocytes and melanocytes in response to APMFAb, we believe that APMFAb may have benefit activities in skin wound healing and whitening.
[References]
- Santoro MM, Gaudino G. Cellular and molecular facets of keratinocyte reepithelization during wound healing. Exp Cell Res. 2005, 304, 274‐286.
- Li J, Chen J, Kirsner R. Pathophysiology of acute wound healing. Clin Dermatol. 2007; 25: 9-18.
- O'Toole EA. Extracellular matrix and keratinocyte migration. Clin Exp Dermatol. 2001; 26: 525 -530.
- Tang L, Sierra JO, Kelly R, Kirsner RS, Li J. Wool-derived keratin stimulates human keratinocyte migration and types IV and VII collagen expression. Exp Dermatol. 2012; 21: 458‐460.
- Qian W, Liu W, Zhu D, Cao Y, Tang A, Gong G, Su H. Natural skin-whitening compounds for the treatment of melanogenesis (Review). Exp Ther Med 2020; 20: 173–185.
- Li, Y.; Huang, J.; Lu, J.; Ding, Y.; Jiang, L.; Hu, S.; Chen, J.; Zeng, Q. The role and mechanism of Asian medicinal plants in treating skin pigmentary disorders. J Ethnopharmacol 2019; 245: 112173.
C2. In the introduction, the transition between the two supposed effects of the extract is brutal (lines 48-50).
(Response) Skin pigmentation is commonly found abnormally after skin injuries such as burns, wound, or laser surgery, and during wound healing responses [1]. Abnormal pigmentation in the skin can be caused by abnormal melanin synthesis or deposition [2]. Epidermal melanin is the main determinant of skin color and protects skin cells from a variety of types of damages including ultraviolet (UV) lights [3]……
These contents have been added to the Introduction section. (P2, L51-53) Thank you.
[References]
- Chadwick S, Heath R, Shah M. Abnormal pigmentation within cutaneous scars: A complication of wound healing. Indian J Plast Surg. 2012; 45: 403-411.
- Rodrigues M, Kosaric N, Bonham CA, Gurtner GC. Wound healing: A cellular perspective. Physiol Rev 2019; 99: 665-706.
- Qian W, Liu W, Zhu D, Cao Y, Tang A, Gong G, Su H. Natural skin-whitening compounds for the treatment of mel-anogenesis (Review). Exp Ther Med 2020; 20:173-185.
C3. Lines 86/123/160: “from 1/10 µg / mL to 100 µg / mL” instead “at 1/10 to 100”?
(Response) We revised it according to the reviewer’s comments. (P2, L89; P4, L126; P5, L165)
C4. The concentration is in µg of dry residue / mL of solvent?
(Response) Yes, it is. APMFAb stock solution was prepared at a concentration of 50 mg (dry APMFAb)/mL (DMSO). The concentrations of APMFAb required for each experiment were prepared by diluting APMFAb stock solution with cell culture medium.
C5. Line 86. The authors say that the concentration of 250 µg / mL is toxic for the cell (Fig.1A). The results of this toxicity are not shown.
(Response) We revised it. (P2, L90)
C6. Line 100: "Violet" spots. The sent paper is in black and white.
(Response) Thank you. We revised it. (P3, L104)
C7. Lines 98 and 104: “Viabilities” and “Recombinant” instead of “Via-bilities” and “Recom-binant”?.
(Response) Thank you. We revised those. (P3, Legend of Fig. 1)
C8. Line 125 "at 10 and 50 µg / mL" instead of “at 10 to 50".
(Response) We revised it. (P4, L130)
C9. Lines 157-174: the authors demonstrate that “the extract elevated type IV collagen synthesis and did not affect type I collagen synthesis”. Can the authors discuss this point? Why collagen IV and not collagne I?
(Response) Type 1 and type IV collagen synthesis in keratinocytes were induced or not depending on the kinds of plant extracts. For example, Patrinia scabiosifolia absolute increased the syntheses of collagen type I and IV in HaCaT cells [1]. Artemisia Montana essential oil induced type IV collagen synthesis, but not type I collagen synthesis in HaCaT cells [2]. Digitaria ciliaris absolute did not affect the syntheses of type IV and Type I collagens HaCaT cells [3]. In the present study, APMFAb upregulated type IV collagen synthesis and did not affect type I collagen synthesis in HaCaT cells. These plant extracts contained different chemical components. These findings indicate that induction of type I and type IV collagen synthesis in keratinocytes may be affected depending on the kinds of plant extracts with different chemical components. Therefore, APMFAb may affect skin wound healing by significantly inducing the biosynthesis of type IV collagen.
[References]
- Hwang DI, Won KJ, Kim DY, Kim HB, Li Y, Lee HM. Chemical composition of Patrinia scabiosifolia flower absolute and its migratory and proliferative activities in human keratinocytes. Chem Biodivers. 2019; 16: e1900252.
- Yoon MS, Won KJ, Kim DY, Hwang DI, Yoon SW, Kim B, Lee HM. Skin regeneration effect and chemical composition of essential oil from Artemisia montana. Nat Prod Commun. 2014; 9:1619-1622.
- Park SM, Won KJ, Hwang DI, Kim DY, Kim HB, Li Y, Lee HM. Potential beneficial effects of Digitaria ciliaris flower absolute on the wound healing-linked activities of fibroblasts and keratinocytes. Planta Med. 2020; 86: 348-355.
C10. lines 214-215. The % are obtained by internal standardization? 100 * (Peak area) / (Total area).
(Response) In the present study, each compound peak area was expressed as a percentage of the total area of the identified compounds as commented by the reviewer.
C11. Lines 212-230: The discussion on the correlation between the composition of the extract and its activity is too short. Moreover, for certain molecules, contrary effects are described in the literature. At what concentrations did these effects occur? The authors conclude that there may be interactions between molecules.
(Response) As commented by the reviewer, Each APMFAb component may induce different responses, including contrary effects, in the same cells and cellular types depending on concentration or cell state. In addition, the interaction between APMFAb components may cause synergistic, additive, and antagonistic effects. However, in the present study, we did not investigate the interaction between APMFAb components. These points associated with interactions between molecules will need to be clearly determined through future studies. Thus, we deleted the conclusion associated with interactions between molecules. Additionally, we added more discussion. (P7, L227-235) Thank you for your constructive comments.

Reviewer 3 Report
This manuscript describes the preparation of a plant extract, determining the chemical composition of the plant extract, and the application of the extract in wound-healing promotion, as well as melanogenesis inhibition. This is a very detailed study that provides the scientific community with new knowledge pertaining to the medicinal properties of Angelica polymorpha and therefore the potential commercial application of the plant extract. Here are some comments/suggestions/recommendations for the authors to consider:
- Title: I would suggest changing the title to ‘Chemical composition and in vitro wound healing promotion and melanogenesis inhibition activities of Angelica polymorpha flower extract’
- Abstract and throughout the manuscript: The plant species name should be written with a lower case, and once mentioned, the plant name can be abbreviated to ‘ polymorpha’
- Abstract and throughout the manuscript: What is meant by ‘absolute extracted’? Since the method of extraction is described in the methodology section, it is recommended that the authors simply refer to the ‘flower extract’.
- Abstract: Please refrain from using abbreviations in the abstract such as ‘WST’, ‘BrdU’, ‘MSH’, ‘MAPK’, ‘MITF’, etc.
- Page 2, line 46: Since the second part of the sentence is related to the first, it is recommended to remove ‘heal’ so that the sentence reads ‘…healing may be delayed, and in severe cases, may not be possible.’
- Page 2, line 51: ‘damages’ should be changed to ‘damage’
- Page 2, line 68: ‘Maxim.’ Is not part of the Latin name of the plant and therefore should not be written in italics. ‘Umbelliferae’ should be written in italics.
- Page 2, line 84: Define ‘WST’
- Page 5, line 179 and throughout the manuscript: All percentage solutions should be clarified by whether it is a v/v, w/v or w/w solution.
- Page 7, line 229: ‘Table 2’ should be ‘Table 1’ and the heading should be more informative, e.g., ‘Chemical compounds identified through GC/MS analysis to be present in the flower extract of polymorpha Maxim.’ or something along those lines. A wide variety of other compounds have been reported to be present in other Angelica species. The fact that only a few compounds were observed may be due to the preparation method, detection method, etc. Many other compounds may be present that could be responsible for the bio-activities observed. Please include a sentence or two regarding extracts from Angelica species reported in literature so as to contextualize the results observed in this study.
Author Response
This manuscript describes the preparation of a plant extract, determining the chemical composition of the plant extract, and the application of the extract in wound-healing promotion, as well as melanogenesis inhibition. This is a very detailed study that provides the scientific community with new knowledge pertaining to the medicinal properties of Angelica polymorpha and therefore the potential commercial application of the plant extract. Here are some comments/suggestions/recommendations for the authors to consider:
Comment 1. Title: I would suggest changing the title to ‘Chemical composition and in vitro wound healing promotion and melanogenesis inhibition activities of Angelica polymorpha flower extract’.
(Response) We thank the reviewer for the valuable and constructive suggest. However, authors, including myself, would like to keep the current title as it is.
C2. Abstract and throughout the manuscript: The plant species name should be written with a lower case, and once mentioned, the plant name can be abbreviated to ‘polymorpha’.
(Response) Thank you. We did it.
C3. Abstract and throughout the manuscript: What is meant by ‘absolute extracted’? Since the method of extraction is described in the methodology section, it is recommended that the authors simply refer to the ‘flower extract’.
(Response) Absolutes are extracted with hexane and ethanol, and contains hydrophobic and volatile (aromatic) components. Additionally, absolutes are known to have their characteristic fragrance and strong volatile properties. Based on the reviewer’s recommendation, we did those.
[References]
- Danh LT, Han LN, Triet NDA, Zhao J, Mammucari R, Foster N. Comparison of chemical composition, antioxidant and antimicrobial activity of Lavender (Lavandula angustifolia L.) essential oils extracted by supercritical CO2, hexane and hydrodistillation. Food Bioprocess Technol. 2013; 6: 3481–3489.
- Rout PK, Misra R, SahooS, Sree A, Rao YR. Extraction of kewda (Pandanus fascicularis Lam.) flowers with hexane: composition of concrete, absolute and wax. Flavour Frag J. 2005; 20: 442-444.
- Baydar H, Kineci S. Scent composition of essential oil, concrete, absolute and hydrosol from Lavandin (Lavandula x intermedia Emeric ex Loisel.). J Essent Oil Bear Plants. 2008; 12: 131-136.
C4. Abstract: Please refrain from using abbreviations in the abstract such as ‘WST’, ‘BrdU’, ‘MSH’, ‘MAPK’, ‘MITF’, etc.
(Response) We revised those.
C5. Page 2, line 46: Since the second part of the sentence is related to the first, it is recommended to remove ‘heal’ so that the sentence reads ‘…healing may be delayed, and in severe cases, may not be possible.
(Response) We deleted it. (P2, L47)
C6. Page 2, line 51: ‘damages’ should be changed to ‘damage’’.
(Response) Thank you. We did it. (P2, L55)
C7. Page 2, line 68: ‘Maxim.’ Is not part of the Latin name of the plant and therefore should not be written in italics. ‘Umbelliferae’ should be written in italics
(Response) Thank you. We revised those. (P2, L72)
C8. Page 2, line 84: Define ‘WST’.
(Response) We did it. (P2, L87)
C9. Page 5, line 179 and throughout the manuscript: All percentage solutions should be clarified by whether it is a v/v, w/v or w/w solution.
(Response) Thank you. We did it according to the reviewer’s comments.
C10. Page 7, line 229: ‘Table 2’ should be ‘Table 1’ and the heading should be more informative, e.g., ‘Chemical compounds identified through GC/MS analysis to be present in the flower extract of polymorpha Maxim.’ or something along those lines. A wide variety of other compounds have been reported to be present in other Angelica species. The fact that only a few compounds were observed may be due to the preparation method, detection method, etc. Many other compounds may be present that could be responsible for the bio-activities observed. Please include a sentence or two regarding extracts from Angelica species reported in literature so as to contextualize the results observed in this study.
(Response) Thank you. Based on the reviewer’s comments, we revised points related to Table 1 and components. (P8, L241-242; P7, L226-235)

Round 2
Reviewer 2 Report
Authors have followed the suggestions; in particular, in the introduction, the transition between the two first paragraphs (lines 53-55) helps to better understand the link between skin healing process and abnormal pigmentation.
In the last paragraph of the introduction, the authors do not explain why they assumed that the active ingredients of Angelica polymorpha have properties on skin healing process and pigmentation. They discuss in the introduction only of anti-ulcer effect and inhibition of the activities of human SH-SY5Y neuroblastoma cells of these compounds. To answer this question (comment1), which concerns the introduction (why test this plant in the context of healing and pigmentation?) the authors describe, in the « Author response to report 1 », their results obtained in the study.
Author Response
Response to reviewer’s comment
Manuscript No: molecules-1405214
Title: Wound healing-promoting and melanogenesis-inhibiting activities of Angelica polymorpha Maxim. flower absolute in vitro and its chemical composition
Authors: Su Yeon Lee et al.
Reviewer 2: Authors have followed the suggestions; in particular, in the introduction, the transition between the two first paragraphs (lines 53-55) helps to better understand the link between skin healing process and abnormal pigmentation.
Comment 1. In the last paragraph of the introduction, the authors do not explain why they assumed that the active ingredients of Angelica polymorpha have properties on skin healing process and pigmentation. They discuss in the introduction only of anti-ulcer effect and inhibition of the activities of human SH-SY5Y neuroblastoma cells of these compounds. To answer this question (comment1), which concerns the introduction (why test this plant in the context of healing and pigmentation?) the authors describe, in the « Author response to report 1 », their results obtained in the study.
(Response) It has been that extracts of Angelina species have the ability to induce proliferation of keratinocytes or inhibit melanin synthesis in melanocytes [1,2]. APM extracts have been known to have anti-ulcer effects in vivo and inhibit the activities of human SH-SY5Y neuroblastoma cells. However, no previous study has investigated the skin wound healing and whitening activities of APM or its crude extract. Thus, in the present study, we investigated the effects of APM flower absolute (APMFAb) on skin wound healing- and whitening-related events in vitro using human immortalized keratinocytes (HaCaT cells) and melanoma cells (B16BL6 cells), respectively, and the mechanisms involved. This issue was described in the Introduction section. (P2, L93-94) Thank you for your constructive coments.
[References]
- Bai, X.; Hu, D.; Wang, Y.; Su, Y.; Zhu, X.; Tang, C. Effects of Angelica dahurica extracts on biological characteristics of hu-man keratinocytes. Zhongguo Xiu Fu Chong Jian Wai Ke Za Zhi 2012, 26, 322-325.
- Park, Y.; Kim, D.; Yang, I.; Choi, B.; Lee, J.W.; Namkoong, S.; Koo, H.J.; Lee, S.R.; Park, M.R.; Lim, H.; Kim, Y.K.; Nam, S.J.; Sohn, E.H. Decursin and Z-ligustilide in Angelica tenuissima root extract fermented by Aspergillus oryzae display anti-pigment activity in melanoma cells. J Microbiol Biotechnol 2018, 28,1061–1067. DOI: 10.4014/jmb.1812.02044.